# Risk Factors for Tooth Loss in Patients with ≥25 Remaining Teeth Undergoing Mid-Long-Term Maintenance: A Retrospective Study

**DOI:** 10.3390/ijerph18137174

**Published:** 2021-07-05

**Authors:** Hiroo Kawahara, Miho Inoue, Kazuo Okura, Masamitsu Oshima, Yoshizo Matsuka

**Affiliations:** 1Department of Stomatognathic Function and Occlusal Reconstruction, Graduate School of Biomedical Sciences, Tokushima University, 3-18-15 Tokushima, Tokushima 770-8504, Japan; hiro32@bronze.ocn.ne.jp (H.K.); inoue.miho@tokushima-u.ac.jp (M.I.); okura.kazuo@tokushima-u.ac.jp (K.O.); m-oshima@tokushima-u.ac.jp (M.O.); 2Kawahara Dental Clinic, 1-128 Muneshige, Mima, Tokushima 771-2104, Japan

**Keywords:** tooth loss, risk factors, occlusal units, non-vital teeth, remaining teeth, posterior load

## Abstract

Tooth loss represents a diffused pathologic condition affecting the worldwide population. Risk factors have been identified in both general features (smoking, diabetes, economic status) and local tooth-related factors (caries, periodontitis). In this retrospective study, we examined the data of 366 patients with a large number of remaining teeth (≥25) undergoing maintenance therapy in order to identify specific risk factors for tooth loss. The number of remaining teeth, number of non-vital teeth, and number of occlusal units were investigated for their correlation with tooth loss. The mean follow-up of patients was 9.2 years (range 5 to 14). Statistically significant risk factors for tooth loss were identified as number of remaining teeth at baseline (*p* = 0.05), number of occlusal units (*p* = 0.03), and number of non-vital teeth in posterior regions (*p* < 0.001). Multiple logistic regression showed that the number of occlusal units and number of non-vital teeth in the posterior regions were significantly associated with a greater risk of tooth loss (odds ratio 1.88 and 3.17, respectively). These results confirm that not only the number of remaining teeth, but also their vital or non-vital status and the distribution between the anterior and posterior regions influence the long-term survival.

## 1. Introduction

Tooth loss significantly affects quality of life and represents a reliable marker of the oral health and socio-economic status of a population [1,2]. Continuous dental education, prevention programs, and treatment advancements significantly reduce tooth loss in the adult population of developed countries. Nevertheless, it still represents one of the 100 top conditions that affects the world’s population [3]. The main causes of tooth loss are dental caries and periodontitis [4,5]. Several studies have been conducted with the aim to define risk factors that predispose individuals to tooth loss. Some patient-related features and tooth-related characteristics were outlined. In particular, age [6], smoking [7], diabetes [6], and educational and economic status [8] have been identified as general risk factors for tooth loss. Tooth-related factors, such as the severity of periodontitis [9], tooth health status (decays, fillings) [9], and the number of remaining teeth [10], have also been proposed.

Dental education, hygiene instructions, and regular professional maintenance have been demonstrated to significantly reduce the risk of tooth loss [11]. However, even among patients who regularly undergo maintenance practices, there is a certain number of teeth that undergo extraction. The level of compliance with maintenance [12], number of residual teeth, their distribution between anterior and posterior areas [10,13], as well as the number of non-vital teeth [14] have been proposed to determine the different rates of tooth loss in patients regularly undergoing dental maintenance visits.

In our previous study, we investigated the general risk factors for tooth loss in patients undergoing mid-to long-term maintenance at a private Japanese general clinic [15]. Among the considered factors (compliance, sex, age, smoking, diabetes, periodontal bone loss, remaining teeth, and use of removable dentures), the number of remaining teeth at the start of maintenance was a statistically significant factor for further tooth loss. Based on these results, we decided to investigate the intrinsic oral factors that may play a role in tooth loss. With the purpose of integrating the findings of our previous study, the aim of this study was to examine only patients with a large number (≥25) of remaining teeth at the start of maintenance, in order to identify specific risk factors for tooth loss. That is, in our previous studies, which were aimed at defining risk factors for tooth loss under maintenance, the number of remaining teeth in sample patients was wide (range 1 to 28). However, in the current study, we only investigated patients with a large number (≥25) of teeth remaining at the start of maintenance. This approach was chosen to exclude the confounding factor of differential occlusal loads existing between patients with large and small number of residual teeth and investigate other specific factors, like tooth vitality and their distribution in the mouth.

## 2. Materials and Methods

### 2.1. Study Design and Patient Sampling

This retrospective cross-sectional study investigated data of 366 patients treated in a Japanese private dental clinic by a general dentist (Hiroo Kawahara) between 2002 and 2011, and who underwent regular maintenance in the same clinic. Written informed consent was obtained from all participants. At the first visit, full-jaw radiography, photographs, and periodontal examinations were performed. The baseline for observation was established after the patients completed their active treatment (tooth extraction, restorative, endodontic, and periodontal therapy) according to individual needs [16,17]. After this active phase of treatment, patients were subjected to reevaluation of clinical parameters (see inclusion criteria). This reassessment phase was used as the baseline for each patient. Education about main oral diseases and associated risk factors was performed, and the importance of their prevention was explained by the general dentist and hygienists. A personalized maintenance protocol was defined based on the patient’s periodontal status at baseline. Briefly, the maintenance protocol comprised periodical clinical examination, professional dental hygiene, and application of fluoride, if needed. Specific interventions, such as tooth extractions, restorations, and endodontic or prosthetic treatments, were performed during maintenance if deemed appropriate by the dentist (Hiroo Kawahara).

All patients underwent the final evaluation at the end of 2016. The duration of maintenance was determined from the baseline of each patient until the end of 2016. For other details about the study design, please refer to the previous manuscript [15].

### 2.2. Inclusion Criteria

The following inclusion criteria were established at baseline:(1)Number of teeth at baseline ≥25;(2)<10% of sites with bleeding on probing;(3)An overall plaque score <15%;(4)<10% of sites with a probing depth of 4 mm;(5)No defective restorations;(6)No active dental caries.

Patients who did not to meet the criteria (2–6) were subjected again to active treatment and reassessed. Patients who underwent implant treatment were excluded. Patients selection process is represented in Figure 1.

### 2.3. Variables and Data Collection

Clinical and radiologic records of the included patients were examined, and the following data were collected: age at baseline, follow-up duration, remaining teeth at baseline, number of non-vital teeth (teeth with non-vital dental pulp; e.g., endodontically treated) in the premolar and molar regions, number of non-vital teeth in the anterior tooth region, and number of occlusal units. The number of occlusal units was defined as follows: any opposing pair of maxillary and mandibular teeth with the same tooth number was counted as one occlusal unit, so that the maximum number of occlusal units in a 28-tooth dentition was 14.

The variables, including the number of remaining teeth, number of non-vital teeth in the premolar and molar region, number of non-vital teeth in the anterior region, and number of occlusal units, were dichotomized based on the median number of data distributions.

### 2.4. Outcomes

The outcome evaluated at the final time point (end of 2016) was the total number of teeth lost during the observational period. The cause of tooth extraction was decided by the dentist and included root fracture, destructive dental caries, severe periodontal impairment, apical lesions, or tooth extraction for convenience. The extraction of wisdom and deciduous teeth was not considered in final calculation.

### 2.5. Statistical Analysis

Logistic regression analysis was performed to determine the relationships between tooth loss and the following factors: remaining teeth at baseline, number of occlusal units at baseline, number of anterior non-vital teeth, and number of non-vital teeth in the premolar and molar regions at baseline. First, logistic regression analysis was performed for each variable, and then multiple logistic regression analysis was performed. The variables at baseline were selected to define the risk factors for tooth loss during maintenance. All statistical analyses were performed using JMP version 14 (SAS Institute, Cary, NC, USA), and a *p*-value < 0.05 indicated statistical significance. Mean values, standard deviations (SD) and confidence intervals (CI) were reported.

### 2.6. Ethics Approval and Consent to Participate

This study was approved by the Clinical Research Ethics Review Committee of Tokushima University Hospital (approval number: 3007-1. Patients’ right to privacy protection was respected, and written informed consent was obtained from all patients. This study was conducted in accordance with the Declaration of Helsinki established by the World Medical Association.

## 3. Results

### 3.1. Participants

A total of 366 patients were deemed eligible for inclusion in the study. Their summary characteristics are listed in Table 1. The average observational period was 9.2 years (range 5 to 14). The distribution of observational periods is presented in Table 2.

### 3.2. Outcome Data

According to the investigated variables, most patients in the study sample (322/366) had 0–8 non-vital teeth in the posterior region. The number of patients by number of non-vital teeth in the premolar and molar regions is shown in Table 3.

Nearly half (180/366) of the patients had no non-vital teeth in the anterior region. The remaining patients had 1–10 non-vital teeth in the anterior region. Their distributions are summarized in Table 4.

The majority of patients (245/366) had 14 occlusal surfaces. The minimum number of occlusal surfaces was 10, which was detected in two patients. The other participants had 11–13 occlusal surfaces. The number of patients according to number of occlusal units is shown in Table 5.

Over the observation period, 198 teeth were lost: 27 teeth were lost due to dental caries, 17 teeth were lost due to periodontal causes, 143 teeth were extracted for root fracture, and 11 teeth were extracted for other causes. Other causes have been described in detail in a previous paper. [15] characteristic study variables and association with tooth loss are shown in Table 6.

Logistic regression analysis was carried out for each variable. There were significant differences between the following variables and the occurrence of tooth loss: remaining teeth at baseline (*p* = 0.05), number of occlusal units at baseline (*p* = 0.03), and number of non-vital teeth in the premolar and molar regions (*p* < 0.001) at baseline.

The results of multiple logistic regression analyses of the study variables with tooth loss showed that the number of occlusal units and number of non-vital teeth in the premolar and molar regions were significantly associated with a greater risk of tooth loss, with odds ratios of 1.88 and 3.17, respectively (Table 7).

## 4. Discussion

In our previous study, various dental and general patient characteristics were investigated for their possible correlation with tooth loss in a cohort of patients undergoing regular maintenance [15]. The results showed that the number of remaining teeth at the start of maintenance was a statistically significant risk factor for further tooth loss. Based on this evidence, we aimed to identify the causes of tooth loss in patients with a large number of remaining teeth (≥25). Furthermore, it was hypothesized that, in addition to the number of remaining teeth, their distribution between the anterior and posterior sectors of the mouth is important, since the chewing load on them is not the same. In particular, we investigated whether the number of non-vital teeth in the anterior sector and the premolar-molar sector correlated with tooth loss. In addition, an assessment was made based on the number of occlusal surfaces available to the patients. Data analysis revealed that the number of occlusal surfaces was strongly correlated with tooth loss. In particular, patients with ≤13 occlusal surfaces had a greater probability of tooth loss than those with 14 occlusal surfaces, with an odds ratio of 1.88. To the best of our knowledge, few studies investigated the influence of occlusal support to the tooth loss. Fushida et al. [13], examined the role of posterior occlusal support in accelerating tooth loss, stating in their conclusions a positive association between the two variables. Sato and collaborators [18] investigated the relationship between occlusal support and tooth loss, finding that a major risk of tooth loss occurs when the remaining number of occlusal contacts is between 5 and 9 sites.

It has been widely demonstrated in the literature that non-vital teeth are more prone to fractures and subsequent tooth extraction [19]. A study by Suzuki et al. [14] showed that the presence of >8 non-vital teeth was correlated with a greater loss of teeth during maintenance. However, due to the wide range of remaining teeth in previously conducted studies, the number of remaining teeth and other factors may have contributed to this. In addition, the non-vital teeth were not divided into anterior and molars. In our study, the range of the number of remaining teeth was narrow, and the anterior and posterior non-vital teeth were distinguished for final evaluation. The results of our observations revealed that the number of non-vital teeth in the premolar–molar region significantly correlates with tooth loss, with an odds ratio of 3.17, for patients with ≥4 endodontically treated teeth, compared to the group of patients with ≤3 non-vital posterior teeth. However, no correlation was observed between tooth loss and the number of anterior non-vital teeth (*p* = 0.82).

Regarding demographic characteristics, the cohort was represented by patients with a mean age of 51.8 years at baseline, and comprised men (38.5%) and women (61.5%). Age and sex were not correlated with tooth loss (*p* = 0.81, *p* = 0.10, respectively). The other results regarding the role of compliance, general health status, smoking, and periodontal bone loss, already presented and discussed in our previous study [15], were also confirmed for this cohort. Briefly, most lost teeth were non-vital teeth and the most common cause of tooth loss was tooth fracture. Other factors like smoking and diabetes were not significantly correlated with tooth loss (*p* = 0.38 and *p* = 0.17, respectively). Regarding periodontal status, the level of bone resorption was evaluated for each tooth and categorized in four classes, subsequently dichotomized for statistical purpose. From statistical analysis no correlation emerged between the level of periodontal bone resorption and tooth loss (*p* = 0.37). Furcation exposure, observed in 96 patients (26%) was not related to tooth loss (*p* = 0.71).

Regarding the generalizability of our results, two important considerations must be made. First, patients were part of the private practice of a general dentist and the periodontal status was healthy-to-moderate for most of the patient population. Few patients had severe periodontitis. Second, all patients were on a maintenance program after being properly instructed and informed about the importance of oral health prevention. A correct use of appropriate tools for home hygiene was promoted [20]. For this reason, few teeth were lost for periodontal reasons (8.5%). These conditions must be considered for the generalizability of the results.

A limitation of this study concerns the lack of consideration of the cause of the loss of devitalized teeth. That is, whether they were lost due to periapical infections or root fractures has not been assessed. Furthermore, it was not part of the objectives of this study to evaluate how non-vital teeth were reconstructed (with direct restorations or prosthetic crowns), and we are aware that this factor could influence the survival of endodontically treated teeth.

The main strength of this study is the long follow-up period (range 5 to 14 years), which allowed us to perform long-term evaluations. Nevertheless, further studies should be conducted to confirm the obtained results and identify other specific risk factors that predispose patients to tooth loss. Furthermore, periodontal status of the included teeth should be considered more specifically, since it could play a big role in tooth stability, resistance to chewing loads and tooth maintenance.

## 5. Conclusions

Considering all of the above reported facts, we conclude that the support of the posterior dentition is important in preventing tooth loss in patients under maintenance. Likewise, the greater the number of non-vital premolars and molars, the greater the possibility of losing teeth in the medium to long term.

## Figures and Tables

**Figure 1 ijerph-18-07174-f001:**
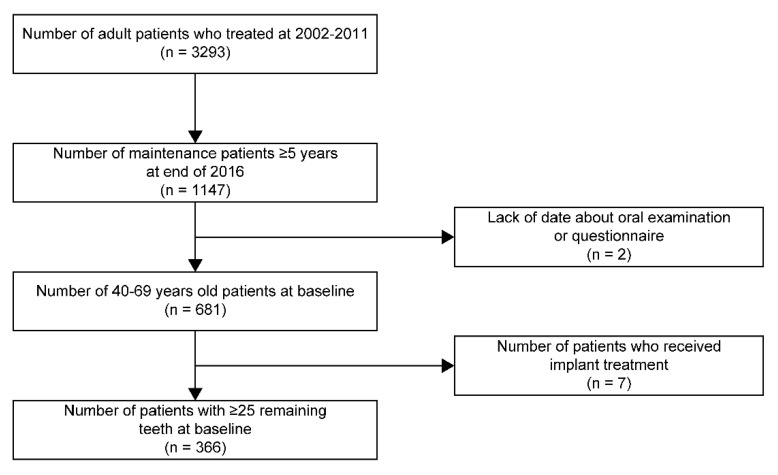
Flowchart of patients selection.

**Table 1 ijerph-18-07174-t001:** Characteristics of patients in this study.

Sample Size (N)	Age at Baseline (Years) Mean ± SD	Year from Baseline (Years) Mean ± SD	Remaining Teeth at Baseline (No. of Teeth) Mean ± SD	No. of Non-Vital Teeth in the Premolar and Molar regions (No. of Teeth) Mean ± SD	No. of Non-Vital Teeth in the Anterior Tooth Regions (No. of Teeth) Mean ± SD	No. of Occlusal Units (N) Mean ± SD	Total No. of Teeth Lost	Tooth Loss/Year per Patients (No. of Teeth)
366	51.8 ± 7.9	9.2 ± 2.6	26.7 ± 1.1	1.5 ± 2.0	4.3 ± 3.4	13.5 ± 3.4	198	0.06

**Table 2 ijerph-18-07174-t002:** Distribution of the observation periods.

Observation Period (Years)	5	6	7	8	9	10	11	12	13	14	Total
**Number**	40	59	35	50	58	31	28	23	19	23	366

**Table 3 ijerph-18-07174-t003:** Number of patients by number of non-vital teeth in the premolar and molar regions.

Number of Non-Vital Teeth in the Premolar and Molar Areas	0	1	2	3	4	5	6	7	8	9	10	11	12	13	14	15	16	Total
**Number of Patients**	35	46	46	45	40	36	25	29	20	10	10	9	7	5	2	0	1	366

**Table 4 ijerph-18-07174-t004:** Number of patients by number of non-vital teeth in the anterior tooth region.

Number of Non-Vital Teeth in the Anterior Tooth Area	0	1	2	3	4	5	6	7	8	9	10	11	12	Total
**Number of patients**	180	61	40	26	20	19	10	4	3	0	3	0	0	366

**Table 5 ijerph-18-07174-t005:** Number of patients by number of occlusal units.

Number of Occlusal Units	11 (4 + 7)	10 (5 + 5)	11 (5 + 6)	12 (5 + 7)	12 (6 + 6)	13 (6 + 7)	14 (7 + 7)
**Number of patients**	3	2	3	27	21	65	245

The numbers in parentheses indicate the rearmost occlusal unit sites on the left and right sides.

**Table 6 ijerph-18-07174-t006:** Relationships of study variables with tooth loss.

		Sample Size (n)	Age at Baseline (Years) Mean ± SD	Year from Baseline (Years) Mean ± SD	Remaining Teeth at Baseline (Teeth Number) Mean ± SD	Tooth Loss (Teeth Number)	Tooth Loss/Year per Patients (Teeth Number)	Logistic Regression Analyses	
								Odds Ratio (95% CI)	*p*-Value
**Sex**	**Male**	141 (38.5%)	51.8 ± 8.0	9.09 ± 2.7	26.6 ± 1.1	93 (47.0%)	0.07	0.69 (0.44–1.08) 0.10	
	**Female**	225 (61.5%)	51.7 ± 7.9	9.19 ± 2.6	26.8 ± 1.1	105 (53.0%)	0.05		
**Age**	**40–54**	230 (62.8%)	46.7 ± 4.4	9.2 ± 2.6	26.8 ± 1.1	113 (57.1%)	0.05	1.06 (0.67–1.66) 0.81	
	**55–69**	136 (37.2%)	60.5 ± 3.9	9.1 ± 2.6	26.5 ± 1.1	85 (42.9%)	0.07		
**No. of remaining teeth**	**26–25**	160 (43.7%)	53.0 ± 7.9	9.3 ± 2.5	25.5 ± 0.5	107 (54.0%)	0.07	0.64 (0.41–1.00) 0.05	
	**28–27**	206 (56.3%)	50.8 ± 7.8	9.1 ± 2.7	27.6 ± 0.5	91 (46.0%)	0.05		
**No. of occlusal units**	**14**	244 (66.7%)	50.5 ± 7.6	9.0 ± 2.6	27.2 ± 0.9	111 (56.1%)	0.05	1.97 (1.26–3.10) 0.003	
	**≤13**	122 (33.3%)	54.5 ± 7.8	9.4 ± 2.6	25.6 ± 0.7	87 (43.6%)	0.08		
**No. of non-vital teeth in the premolar and molar regions**	**≤3**	172 (47.0%)	51.5 ± 7.8	9.2 ± 2.6	26.9 ± 1.1	51 (25.8%)	0.03	3.31 (2.08–5.29) ≤0.001	
	**≥4**	194 (53.0%)	52.1 ± 8.0	9.1 ± 2.6	26.5 ± 1.1	147 (74.2%)	0.08		
**No. of non-vital teeth in the anterior tooth regions**	**0**	180 (49.2%)	51.8 ± 7.7	9.4 ± 2.7	27.0 ± 1.1	91 (46.0%)	0.05	1.05 (0.68–1.62) 0.82	
	**≥1**	186 (50.8%)	51.8 ± 8.1	8.9 ± 2.5	26.5 ± 1.1	107 (54.0%)	0.06		

**Table 7 ijerph-18-07174-t007:** Multiple logistic regression analyses of study variables with tooth loss.

		Coefficient	Standard Error	χ^2^	Odds Ratio (95% CI)	*p*-Value
**No. of remaining teeth**	26–25	−0.08	0.30	0.07	1	0.79
	28–27				0.92 (0.51–1.67)	
**No. of occlusal units**	14	0.63	0.31	4.12	1	0.04
	≤13				1.88 (1.02–3.48)	
**No. of non-vital teeth in the premolar and molar regions**	≤3	0.57	0.12	23.03	1	≤0.001
	≥4				3.17 (1.98–5.09)	

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
