# Peer review of "Risk Factors for Tooth Loss in Patients with ≥25 Remaining Teeth Undergoing Mid-Long-Term Maintenance: A Retrospective Study"

_ijerph, 2021, doi:10.3390/ijerph18137174_

Round 1
Reviewer 1 Report
Manuscript ID: ijerph-1272136 This manuscript, Risk Factors for Tooth Loss in Patients with ≥ 25 Remaining Teeth Undergoing Mid-Long-Term Maintenance: A Retrospective Study, is an interesting paper regarding examination of the risk factors for tooth loss for only patients with a large number of remaining teeth. The paper is well-thought and should be of interest to IJERPH readers. Moreover, this paper is a continuation of the previous article (doi.org/10.3390/ijerph17176258). However, there are some points that should be considered and/or improved: Abstract: [1] You stated ,,in the current study, we only investigated patients with a large number of teeth remaining at the start of maintenance". Please emphasize the benefits of such a retrospective study compared to previous ones. Materials and Methods: [2] Please explain the assumed age criterion (40-69 years old).Results: [3] In Table 3 please add unit (years).
[4] Remove the abbreviations (SD/CI) information from the Table and put them in section 2.5. Statistical analysis. [5] In Tab. 6 the last column (Logistic regression analyses is difficult to read and requires formatting. Discussion: [6] You state: ,,The other results regarding the role of compliance, general health status, smoking, and periodontal bone loss, already presented and discussed in our previous study, were also confirmed for this cohort." It would be useful for potential readers to also, here, briefly familiarize the results, do not just refer to the previous article.
[7] The Discussion seems incomplete, there is no reference to other scientific publications concerning such or similar retrospective studies. Also, the results should be compared with the results from other clinics - this allows more general conclusions to be drawn. Conclusions: [8] Your statement: ,,Further studies should be conducted to confirm the obtained results and identify other specific risk factors that predispose patients to tooth loss" should be transferred to the Discussion and also expanded.
Author Response
Response to Reviewer’s Comments
Reviewer 1 Comments to Author:
- You stated, in the current study, we only investigated patients with a large number of teeth remaining at the start of maintenance". Please emphasize the benefits of such a retrospective study compared to previous ones.
Reply:
Thank you for your comments. We added a clarification in Introduction paragraph, explaining the rational of such choice.
This approach was chosen to exclude the confounding factor of differential occlusal loads existing between patients with large and small number of residual teeth and investigate other specific factors, like tooth vitality and their distribution in the mouth.
- Materials and Methods: [2] Please explain the assumed age criterion (40-69 years old).
Reply:
Thank you for your comment. We checked the protocol and it was not a criterion of inclusion, but only a descriptive data of included population. So, it has been removed from the inclusion criteria.
- In Table 3 please add unit (years).
Reply:
Thank you for your comment. Unit has been added.
- Remove the abbreviations (SD/CI) information from the Table and put them in section 2.5.
Reply:
Thank you for your comment. The abbreviations have been removed from the Tables footnotes and added in paragraph 2.5.
Mean values, Standard Deviations (SD) and Confidence intervals (CI) are reported in Tables 1, 6 and 7.
- In Tab. 6 the last column (Logistic regression analyses is difficult to read and requires formatting
Reply:
Thank you for your comment. The table has been formatted.
- You state: ,,The other results regarding the role of compliance, general health status, smoking, and periodontal bone loss, already presented and discussed in our previous study, were also confirmed for this cohort." It would be useful for potential readers to also, here, briefly familiarize the results, do not just refer to the previous article.
Reply:
Thank you for your comment. A brief summary of the results have been added to the discussion paragraph.
Briefly, most lost teeth were non-vital teeth and the most common cause of tooth loss was tooth fracture. Other factors like smoking and diabetes resulted to be not significantly correlated with tooth loss.
- The Discussion seems incomplete, there is no reference to other scientific publications concerning such or similar retrospective studies. Also, the results should be compared with the results from other clinics - this allows more general conclusions to be drawn.
Reply:
Thank you for your comment. A paragraph discussing findings of similar studies has been added.
To the best of our knowledge, only few studies investigated the influence of occlusal support to the tooth loss. Fushida et al. [13], examined the role of posterior occlusal support in accelerating tooth loss, stating in their conclusions a positive association be-tween the two variables. Sato and collaborators [18] investigated the relationship between occlusal support and tooth loss, finding that a major risk of tooth loss occurs when the remaining number of occlusal contacts is between 5 and 9 sites.
- Conclusions: Your statement: ,,Further studies should be conducted to confirm the obtained results and identify other specific risk factors that predispose patients to tooth loss" should be transferred to the Discussion and also expanded.
Reply:
Thank you for your comment. The statement has been transferred to the Discussion and expanded.
Nevertheless, further studies should be conducted to confirm the obtained results and identify other specific risk factors that predispose patients to tooth loss. Furthermore, periodontal status of the included teeth should be considered more specifically, since it could play a big role in tooth stability, resistance to chewing loads and tooth maintenance.
Reviewer 2 Report
This manuscript reports an interesting study. It is well written and after some relatively minor revisions, I would be happy to recommend that it is accepted for publication. These revisions are:
- Since the advent of the American Academy for Periodontology (AAP)/European Federation for Preiodontology (EFP) classification for periodontal and peri-implant diseases (Caton et al. 2018) the term "periodontal disease" is obsolete and should now longer be used. Throughout the manuscript, the word"periodontitis" should be used to replace the word "periodontal disease. "
- The page numbers in references 3,4,8,12, 18 should be written in full e.g for reference 3 as 592 - 597 and not as 592-7.
- References 2, 6, 9, 10, 11, 15 appear to have either incomplete or no page numbers. This may be because some relate to e-publications. However, not all of the journals concerned appear to be e-publications.
Author Response
Response to Reviewer’s Comments
Reviewer 2 Comments to Author:
- Since the advent of the American Academy for Periodontology (AAP)/European Federation for Preiodontology (EFP) classification for periodontal and peri-implant diseases (Caton et al. 2018) the term "periodontal disease" is obsolete and should now longer be used. Throughout the manuscript, the word"periodontitis" should be used to replace the word "periodontal disease. "
Reply:
Thank you for your comment. The corrections have been performed.
- The page numbers in references 3,4,8,12, 18 should be written in full e.g for reference 3 as 592 - 597 and not as 592-7.
Reply:
Thank you for your comment. References have been corrected.
- References 2, 6, 9, 10, 11, 15 appear to have either incomplete or no page numbers. This may be because some relate to e-publications. However, not all of the journals concerned appear to be e-publications.
Reply:
Thank you for your comment. We verified each reference, but pages are available only for N°15. The references 10 and 11 appear to have page numbers: 105 and 328 respectively. No further information about page numbers are available.
Reviewer 3 Report
Manuscript not eligible for publication, the authors do not offer new perspectives compared to those already known and affirm it themselves in the conclusions.
Author Response
Response to Reviewer’s Comments
Reviewer 3 Comments to Author:
Manuscript not eligible for publication, the authors do not offer new perspectives compared to those already known and affirm it themselves in the conclusions.
Reply:
Dear Reviewer, thank you for the time and consideration you dedicated to our manuscript. We appreciate your opinion, nevertheless, we should make note that some aspects investigated in this study are firstly presented in the literature. In particular, previous studies have shown that under maintenance, more non-vital teeth result in more tooth loss. However, due to the wide range of remaining teeth in such studies, the number of remaining teeth and other factors may have contributed to this. In addition, the non-vital teeth were not divided into anterior and posterior. This time, the range of the number of remaining teeth was narrow, making possible to exclude confounding factors, such as different chewing load, and the anterior and posterior non-vital teeth were separately evaluated for association with tooth loss.
Round 2
Reviewer 1 Report
Manuscript ID: ijerph-1272136 This manuscript, Risk Factors for Tooth Loss in Patients with ≥ 25 Remaining Teeth Undergoing Mid-Long-Term Maintenance: A Retrospective Study, is an interesting paper regarding examination of the risk factors for tooth loss for only patients with a large number of remaining teeth. The paper is well-thought, is a continuation of the previous article (doi.org/10.3390/ijerph17176258), and also should be of interest to IJERPH readers. Moreover, the manuscript has undergone significant improvements during the review process, and now, I would like to recommend the article for acceptance in present form.Author Response
Please see the attachment.

Reviewer 3 Report
Thank you for your reply.
I had the opportunity to read the revised manuscript, I must say that it has improved significantly, I would like to add some observations:
• Add in the discussions in addition to smoking as a risk factor, a motivational plan to reduce the occurrence of periodontitis through the home use of electric and sonic toothbrushes, I am attaching a reference:
The efficacy of powered oscillating heads vs. Powered sonic action heads toothbrushes to maintain periodontal and peri-implant health: A narrative review, , , , ,
Author Response
Reviewer 3 Comments to Author
- Add in the discussions in addition to smoking as a risk factor, a motivational plan to reduce the occurrence of periodontitis through the home use of electric and sonic toothbrushes, I am attaching a reference:
The efficacy of powered oscillating heads vs. Powered sonic action heads toothbrushes to maintain periodontal and peri-implant health: A narrative review, International Journal of Environmental Research and Public Health, 2021, 18(4), pp. 1–17, 1468
Reply:
Thank you for your comment. A reference to the importance of home hygiene tools has been added to the discussion. “A correct use of appropriate tools for home hygiene was promoted. [Ref 20]”
- In the posterior elements with bone loss, did they have exposed furcations?
Reply: Thank you for your comment. Yes, furcation exposure was present in 96 patients (26%), but it resulted to be not related to tooth loss (p=0.71). A paragraph with this information has been added to the discussion.